# Priority on High-Quality: Instruction Data Selection for Optimized Instruction Tuning

## Abstract

Large Language Models (LLMs) have demonstrated a remarkable understanding of language nuances through instruction tuning, enabling them to effectively tackle various natural language processing tasks. Previous research on instruction tuning mainly focused on the quantity of instruction data. Recent studies indicate that the quality of instruction data is more significant than the quantity of data. Even selecting a small amount of high-quality data can achieve optimal fine-tuning effects. However, existing selection methods have severe limitations in defining the quality of each instruction data and considering the balance between data quality and data diversity. To address these challenges, we propose a strategy that utilizes noise injection to identify the quality of instruction data. We also implement the strategy of combining inter-class diversity and intra-class diversity to improve model performance. Experimental results demonstrate that our method significantly outperforms the model trained on the full dataset when utilizing only 12% of the entire dataset. Our study provides a new perspective on noise injection in the field of instruction tuning, and also illustrates that a high-quality instruction dataset should possess both quality and diversity. Additionally, we have published our selected high-quality instruction data.

## 1 Introduction

Large Language Models (LLMs) have the ability to carry out intricate natural language processing tasks in various situations and fields through instruction tuning (OpenAI, 2023; Touvron et al., 2023; Caruccio et al., 2024; Chen et al., 2023c; Sun et al., 2023; Ouyang et al., 2022; Iyer et al., 2022). In the realm of instruction tuning, previous researches have primarily concentrated on how the quantity of instruction data impacts training results (Wei et al., 2022; Chung et al., 2022; Longpre et al., 2023). Consequently, some researches focus on researching methods to automatically generate instruction data (Wang et al., 2023; Taori et al., 2023; Xu et al., 2023b), thus promoting the continuous expansion of the scale of instruction data. Training models on constantly expanding datasets is not practical because of the significant costs involved.

Therefore, current researches are investing in research on the quality of instruction data (Zhou et al., 2023; Köpf et al., 2023; Li et al., 2023b). Specifically, LIMA (Zhou et al., 2023) has the potential to enhance the model's ability to track instructions effectively with just 1,000 curated high-quality instruction data. This demonstrates the importance of data quality over data quantity, while also raising the question of how to evaluate the quality of each instruction. Subsequently, Alpagasus (Chen et al., 2023b) uses the external model GPT-3.5-Turbo to score each data and chooses the one with the highest score as a high-quality dataset. The Q2Q (Li et al., 2023a) calculates data quality by instructing fine-tuned model and specific formulas. Assessing with external models fails to consider the pre-trained model's own data preferences.

Simultaneously, a number of researchers adopt a diversity-oriented approach when investigating the nature of high-quality data. LTD (Chen et al., 2023a) retrieves core samples for each type of task from the task data set, and uses these core samples to form a more representative but smaller subset to train the model. Self-Evolved (Wu et al., 2023) uses K-center to enhance the diversity of data. These studies focus too much on the diversity of the model and ignore the efficiency of each piece of data quality, which may lead to a decrease in model performance. Therefore, when selecting high-

quality datasets, what should be considered is an effective combination of data quality and overall diversity.

In this work, we aim to establish a selection method for high-quality data. This involves assessing the quality of each data from the viewpoint of the PLM, while thoroughly contemplating the amalgamation of both quality and diversity. Inspired by previous research on noise utilization (Namysl et al., 2020; Hua et al., 2022; Jain et al., 2023), we propose to define the quality of each data by introducing noise. Specifically, we inject noise into the input part of the instruction, then analyze the changes in the output probability distribution of the pre-trained model for the entire instruction, and select those data with high probability distribution consistency as high-quality data. Moreover, we combine the strategies of inter-class diversity and intra-class diversity to improve the coverage of the selected data and reduce the redundancy in the data set.

In summary, our main contributions are as follows:

- We propose a method for selecting high-quality instruction data without using additional models and taking into account an effective combination of quality and diversity.
- Our method creatively applies noise injection to measure the quality of each instruction data, providing a new application perspective for noise in the field of instruction tuning.
- The overall performance of our method surpasses that of full-data training when selecting 12% of the entire dataset, which not only reduces the training cost, but also improves the performance of the model.
- We publish a high-quality instruction dataset filtered from Alpaca by our proposed method.

## 2 METHOD

### 2.1 MOTIVATION

The study by LIMA (Zhou et al., 2023) indicates that the pre-training phase is where large models accumulate most of their knowledge. In contrast, the goal of instruction tuning is to steer the model towards a particular interaction style or format, effectively demonstrating its built-in knowledge and abilities. From this insight, we formulate a hypothesis: instructions that align with the knowledge absorbed during pre-training are more easily learned and integrated by the model through subsequent fine-tuning. We term these effective guiding instructions as "high-quality instructions."

Identifying high-quality instructions from a vast array of datas has emerged as a pivotal challenge that requires resolution. The smoothness assumption and clustering assumption suggest that data points with different labels are likely to be separated in regions of low density, whereas data points that are similar will exhibit consistency in the model's output (Zhang et al., 2023; Jeong & Shin, 2020; Ouali et al., 2020). This concept leads us to hypothesize that for large language models (LLMs), if the knowledge associated with an instruction has been internalized during pre-training, the model's responses should remain relatively consistent when the instruction is slightly altered, indicating a level of stability.

In our study, we introduce a method grounded in the previously mentioned assumptions. This method involves introducing noise into the low-dimensional embedding space of instructions to generate perturbations, and subsequently tracking the consistency of the model's output responses. We contend that data demonstrating high output consistency provides a clearer indication of the model's learned capabilities, which we utilize as a metric for instruction quality. To prevent data selection bias and its constraints on showcasing the model's abilities, we utilize the k-means clustering algorithm to ensure a varied representation across different categories. Within each cluster, we further enrich the sample diversity by calculating the cosine similarity between data points. The comprehensive methodological framework of this research is detailed in Figure 1.

### 2.2 CONSISTENCY SELECTION

The process of noise injection involves introducing a specific level of disturbance into the instruction data. Adding interference directly to a high-dimensional space such as the original text can easily cause semantic changes. Therefore we perform noise injection on the embedding of the input part

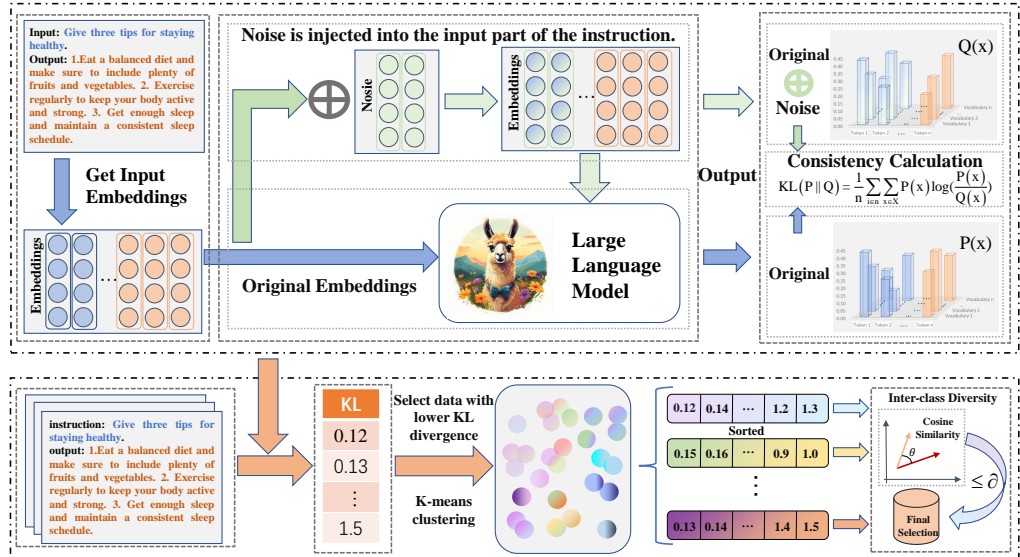

Figure 1: The overall framework. The top portion of the figure illustrates the method for determining the quality of each data, whereas the bottom part depicts the procedure for integrating quality with diversity selection strategies.

of the text. And we use Gaussian noise which is widely used in image processing. In particular, we introduced $\beta$ to change the mean and variance to control the size of the noise. For each instruction $d_i$ in the initial dataset $D_0$, where $d_i$ is represented as $(X, Y)$. The embedding for each $d_i$ instruction is expressed as $(\mathbf{e^x_{1,i}} \cdots \mathbf{e^x_{n,i}}, \mathbf{e^y_{1,i}} \cdots \mathbf{e^y_{m,i}})$. We introduce a specific level of noise to the embedding of the input section of the instructions, as per the following formulas:

$$\mathbf{n_{k,i}} = \beta(\mu^i_{\mathbf{x}} + \sigma^i_{\mathbf{x}}\epsilon_{\mathbf{i}}), \epsilon_{\mathbf{i}} \sim \mathcal{N}(\mathbf{0}, \mathbf{1}), \tag{1}$$

$$\tilde{\mathbf{e}}^{\mathbf{x}}_{\mathbf{k,i}} = \mathbf{e}^{\mathbf{x}}_{\mathbf{k,i}} + \mathbf{n_{k,i}} \tag{2}$$

where $\beta$ represents the scaling factor of noise magnitude, $\sigma^i_x$ denotes the standard deviation of input part $X$ in the $i^{th}$ instruction, and $\mu^i_x$ stands for the mean of input part $X$ in the $i^{th}$ instruction, $\mathbf{e^x_{k,i}}$ represents the embedding of the $\mathbf{k^{th}}$ token in the $i^{th}$ data, $\tilde{\mathbf{e}}^{\mathbf{x}}_{\mathbf{k,i}}$ represents the embedding $\mathbf{e^x_{k,i}}$ after adding noise.

In order to assess the consistency of the model in predicting word-level granularity before and after introducing noise, we collected the probability distribution predictions of the model at each vocabulary position after adding noise. Subsequently, we compared the consistency of model prediction probabilities between the original instructions and the instructions after noise was added. A higher level of consistency indicates better data quality. The formula for calculating the consistency of probabilities is as follows:

$$D_{\mathrm{KL}}(P||Q) = \frac{1}{n} \sum_i P(i) \log\left(\frac{P(i)}{Q(i)}\right) \tag{3}$$

where $n$ represents the token length of an instruction, including the input x and the output y. $P_i$ represents the probability output of the $i_{th}$ instruction after passing through the model, while $Q_i$ denotes the probability output of the $i_{th}$ instruction after adding noise to the input portion and passing through the model.

A lower KL divergence value suggests a greater consistency in the probability distribution, thereby indirectly indicating the quality of the data. when perturbations are introduced, there will be a certain degree of randomness in the actual noise generation. Therefore, in the actual experimental operation, we took three independent sampling processes and calculated the corresponding KL divergence values, and finally took the average of the three as our consistency evaluation result.

## 2.3 DIVERSITY SELECTION

In the previous steps, we quantified the quality of each piece of data through consistency calculations. However, relying solely on consistency calculations for sorting and selection may result in the selected data set having only a few categories, resulting in reduced model performance. In order to improve the category diversity of the selected data set, we adopt the inter-class diversity selection and intra-class diversity selection strategies.

In the inter-class diversity selection strategy, our core goal is to expand the coverage of the selected data while ensuring the quality of each piece of data. To this end, we prioritize data that ranks higher in the initial ranking, while implementing inter-class diversity selection to ensure that the selected data set is broadly representative at the class level. We calculate the overall semantic embedding of each data point using the following formula. We then utilize the K-means (Lloyd, 1982) clustering algorithm for inter-class diversity filtering to optimize the quality of the dataset and the generalization performance of the model. The relevant calculation formulas are as follows:

$$[\mathbf{h_{1,i}^x} \cdots \mathbf{h_{n,i}^x}, \mathbf{h_{1,i}^y} \cdots \mathbf{h_{m,i}^y}] = PLM(\mathbf{e_{1,i}^x} \cdots \mathbf{e_{n,i}^x}, \mathbf{e_{1,i}^y} \cdots \mathbf{e_{m,i}^y}), \tag{4}$$

$$\mathbf{H_i} = \frac{\sum_{k=1}^{n} \mathbf{h_{k,i}^x} + \sum_{k=1}^{m} \mathbf{h_{k,i}^y}}{n+m}, \tag{5}$$

$$(\text{cluster}_1 \cdots \text{cluser}_k) = \text{K-means}(\mathbf{H_1} \cdots \mathbf{H_i}) \tag{6}$$

where $PLM$ denotes a pre-trained model, while $\mathbf{h_{n,i}^x}$ and $\mathbf{h_{m,i}^x}$ indicate the ultimate hidden states of the $i^{th}$ instructions. $H_i$ represents the vector representation of the entire statement.

After confirming data coverage using the inter-class diversity selection strategy, we observed that data points within the same class might exhibit significant similarities, leading to redundant data.To diminish redundancy and enhance dataset diversity, we implemented an intra-class diversity selection strategy. More precisely, we assess the quality of data within each category and then calculate the cosine similarity between instructions by utilizing sentence embedding. The diversity of the dataset is improved by choosing instructions that have similarities under a set limit and adding these less similar data points to the filtered subset.

## 3 EXPERIMENTS

### 3.1 EXPERIMENTAL SETUP

**Datasets** Our filtering object uses the Alpaca (Taori et al., 2023) dataset created by Stanford University, which contains 52K instruction data. To thoroughly assess the model's performance, we utilized a range of datasets for conducting specific capability tests. We use the MMLU (Hendrycks et al., 2021) dataset to measure the model's ability to handle interdisciplinary knowledge in a multilingual environment. By employing the Humaneval (Chen et al., 2021), we evaluate the model's proficiency in comprehending and producing code. The GSM-8K (Cobbe et al., 2021) is utilized to assess the model's aptitude in resolving mathematical problems. In addition, we use the CommonsenseQA (Talmor et al., 2019) to examine the model's mastery of common sense knowledge in daily life. Finally, through the NaturalQuestions (Kwiatkowski et al., 2019), we evaluate the model's performance in understanding and answering questions involving world knowledge.

**Baselines** In this study, we compare various baseline methods. Alpaca-all (Taori et al., 2023) is directly trained on the complete Alpaca dataset.Random is selected from the source data set through random sampling. LIMA (Zhou et al., 2023) is trained on 1k high-quality instruction-following data meticulously handcrafted. AlpaGasus (Chen et al., 2023b) uses ChatGPT to score each piece of data and select the high-scoring data for training. Q2Q (Li et al., 2023a) trains a model initially with a few instructions, and subsequently assess the data quality using two distinct loss values within the model. Additionally we use the length of the instruction's output as a strong baseline (Zhao et al., 2024).

**Implementation Details** We use the Llama-2 (Touvron et al., 2023) model with 7B parameters as the base language model. During training, we fine-tune the model for 3 epochs, with the batch size of 256. We utilize the AdamW optimization algorithm with a learning rate set to $2 \times 10^{-5}$.

Table 1: The overall results on various abilities. "Math" means GSM-8K,"Code" means Humaneval, "World Knowledge" means NaturalQuestions.

| | MMLU | Math | Code | Commonsense | World Knowledge | Average | Δ |
|---|---|---|---|---|---|---|---|
| Alpaca-All | 47.93 | 13.12 | 13.41 | 55.04 | 20.83 | 30.07 | - |
| LIMA | 40.76 | 19.33 | 15.24 | 44.72 | 11.83 | 26.38 | -3.69 |
| Q2Q | 44.69 | 13.5 | 15.85 | 47.75 | 28.84 | 30.13 | +0.05 |
| AlpaGasus | 46.51 | 7.73 | 14.63 | 54.05 | 29.75 | 30.53 | +0.46 |
| Length | 45.87 | 16.07 | 14.02 | 50.07 | 30.66 | 31.34 | +1.27 |
| Random | 45.97 | 10.99 | 11.59 | 52.66 | 29.14 | 30.07 | 0 |
| Ours | 47.12 | 15.69 | 15.85 | 56.51 | 29.83 | 33.00 | **+2.93** |

To enhance the model's performance, we extend the maximum length of input sentences to 4096 tokens. For testing the various capabilities of the model, we use the Opencompass (Contributors, 2023) framework. For MMLU, we utilize 5-shots, and for CommonsenseQA, we use 8-shots. When it comes to multiple-choice questions, we base our judgment on the first letter of the answer provided by the LLMs. Additional implementation details can be found in Appendix A.

## 3.2 MAIN RESULTS

**Changes in Performance** We conduct an in-depth exploration of the data filtering effect under different noise intensities. Specifically, we select 5%-15% of the original dataset as subsets under noise levels of $\beta = 1$ and $\beta = 10$, respectively, and train models based on these subsets. The experimental results are shown in Figure 2. The model trained with the filtered subset generally outperforms the results of training with the full dataset under the two noise intensities, confirming the effectiveness of our proposed approach. Especially under the condition of $\beta = 10$ and a 12% selection ratio, the model performance reaches the optimal level. Additionally, we observe an overall trend toward better model performance at higher noise levels, which may be due to the fact that low noise intensity is not sufficient to cause effective interference in the data. The parameters related to noise injection can be found in the Appendix A.

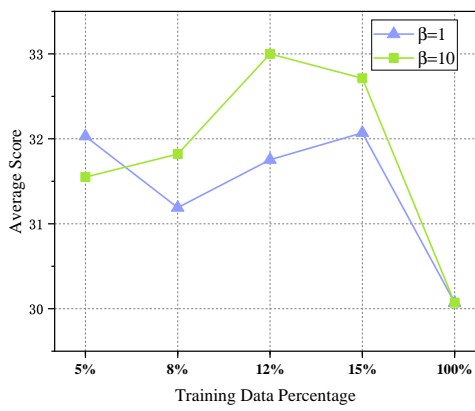

Figure 2: Compare various dataset sizes within the alpaca dataset to assess how our method's performance varies.

**Baseline Comparison** We compare the peak performance of our method with established benchmark methods and some intuitive filtering methods used as baselines. The experimental results are shown in Table 1. Our method, using only about 12% of the Alpaca data, outperforms all results from full Alpaca data training in overall performance and exceeds existing baselines. In the MMLU test, our method is slightly below the results of full data training but notably improves other aspects of the model's capabilities. LIMA significantly outperforms current methods in the Math ability test. This may be due to the extremely long length of each instruction, which makes it easier for the model to generate a more suitable chain-of-thought process. However, focusing solely on length has led to the degradation of other abilities, such as world knowledge, which is significantly lower than various benchmarks. Relying solely on external models for selection without considering their biases may limit the model's performance in specific areas. In particular, AlpaGasus achieves only half the scores of other baselines in terms of mathematical ability.

### 3.3 Generalization of Method

**Different Datasets** Our method demonstrates outstanding performance on the Alpaca dataset, which is bootstrapped from powerful LLMs. To assess whether our method retains its efficacy across different dataset types, we broadened our experimental scope. We chose two distinct datasets for testing: the manually crafted instruction dataset Dolly (Dolly, 2023) and the conventional NLP-related dataset FLAN (Longpre et al., 2023). We applied our filtering method to these datasets and evaluated the performance of the filtered subsets on various test sets. The results are presented in Figure 3. The subsets we selected consistently outperformed the full dataset training. These findings confirm that our method is not only suitable for the Alpaca dataset but also effectively generalizes to other dataset types, facilitating the identification of high-quality instruction data. Further details are provided in Appendix A.

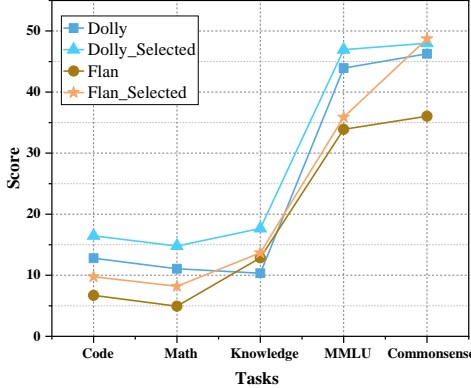

Figure 3: We randomly select a subset from the FLAN dataset that is comparable in size to the Dolly dataset for experiments. we employed the PPL loss as the metric to assess performance in the multiple-choice tests conducted on the Flan dataset.

**Different Models** In our preliminary research, we select the large-scale model Llama2-7B for our experiments to verify the effectiveness and feasibility of our proposed method. To understand the performance of our method across models with varying sizes and parameter configurations, we expand the scope of our experiments. We conduct detailed supplementary tests on two versions of the Qwen2 (Yang et al., 2024) model series, the Qwen2-0.5B and Qwen2-1.5B models. As illustrated in the Table 2, our method's performance within the Qwen2 model series is notable. It not only demonstrates excellent performance across different model sizes but also significantly outperforms the benchmark methods widely recognized in the industry on multiple evaluation metrics. These results suggest that our method can accurately identify high-quality data that aligns with the unique characteristics of the models, be it in the smaller-scale Qwen2-0.5B model or the larger-scale Qwen2-1.5B model.

Table 2: Experiments were conducted on two models of different scales in Qwen2, aiming to verify the generalization capability of our model when faced with different models.

|  | Qwen2 | MMLU | Math | Code | Commonsense | World Knowledge | Average | Δ |
|---|---|---|---|---|---|---|---|---|
| Alpaca-all | 0.5B | 35.83 | 14.56 | 20.73 | 52.01 | 7.59 | 26.14 | — |
| AlpaGasus | 0.5B | 36.23 | 27.22 | 23.17 | 51.92 | 6.54 | 29.02 | +2.88 |
| Ours(14%) | 0.5B | 36.68 | 34.85 | 26.83 | 53.32 | 7.01 | 31.74 | **+5.60** |
| Alpaca-all | 1.5B | 50.47 | 39.73 | 33.54 | 69.94 | 13.77 | 41.19 | — |
| AlpaGasus | 1.5B | 35.59 | 53.98 | 36.59 | 71.25 | 13.77 | 42.24 | +1.05 |
| Ours(15%) | 1.5B | 45.10 | 57.54 | 40.24 | 71.25 | 14.16 | 45.66 | **+4.47** |

### 3.4 Effect of Noise

The cornerstone of our method is the strategic introduction of noise in the data selection phase to pinpoint high-quality noise samples. We replace the conventional Gaussian noise with uniform noise to investigate the impact on model performance. The findings are presented in the Figure 4. The figure clearly illustrates that, across various noise levels, Gaussian noise yields significantly superior experimental outcomes compared to uniform noise. A meticulous comparison of the images within the figure reveals a notable trend: as noise intensity rises, both methods exhibit considerable performance gains. Our research concludes that a moderate increase in noise intensity aids in refining

the identification of data quality. This effect might stem from the fact that moderate noise levels effectively accentuate key data features while diminishing the relevance of less critical details, thus enhancing the efficiency of data quality differentiation.

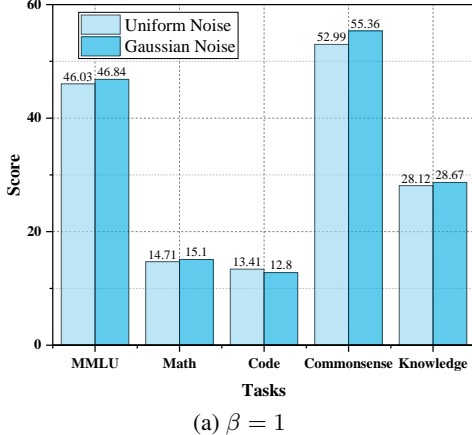

(a) $\beta = 1$

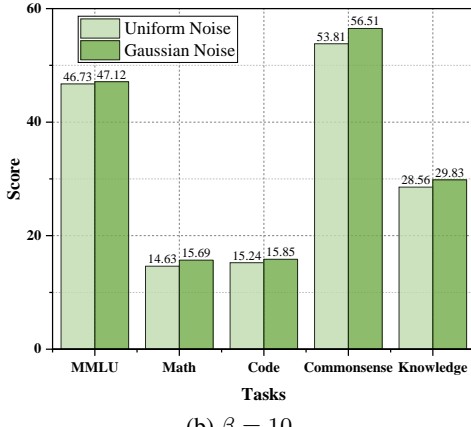

(b) $\beta = 10$

Figure 4: Examining the impacts of varying noise types and varying noise intensities on experimental outcomes. In the figure on the left, $\beta = 1$ signifies the initial intensity of the noise. Conversely, in the figure on the right, $\beta = 10$ suggests a tenfold increase in the noise intensity. In the case of uniform noise, we modulate the intensity of the noise by symmetrically expanding the upper and lower boundaries of the sampled values.

### 3.5 Ablation Experiments

**Consistency** To validate our proposed hypothesis, we intentionally select a dataset with low consistency for training to assess its effect on model performance. During the data screening phase, we prioritize consistency as the sole criterion, omitting additional diversity filters. In the performance evaluation, we not only test the model's overall capabilities but also incorporate the Vicuna test set vicuna2023 from open-domain questions into our analysis. The comprehensive experimental results are detailed in Table 3. The evaluation reveals a pronounced trend: models trained on high-consistency data significantly surpass those trained with other methods. Notably, the performance of models trained on low-consistency data does not achieve even the level of random selection. This result indirectly supports our hypothesis, indicating that high-consistency data is more effective in uncovering the knowledge the model gains during pre-training. Some examples of high and low consistency instructions can be found in the Appendix C.

To delve into this phenomenon, we examine the Vicuna-test results and randomly select two questions to assess the impact of various selection methods. The comparative outcomes are detailed in Figure 5. Our analysis reveals that models trained on highly consistent data exhibit significant benefits in question-answering. These models produce answers that are not only richer in content but also more fluent in language, while maintaining better contextual coherence. This discovery underscores the critical role of selecting consistent data for training. Utilizing high-consistency data ensures that the model rapidly adapts to a unified data style and format, enhancing the effective application and demonstration of the knowledge absorbed during pre-training.

Table 3: The comparison of consistency selection experiments. "Ability" refers to the collective mean of diverse testing capabilities. For the Vicuna test, we utilize "weighted_alpaca_eval_gpt4_turbo" from AlpacaEval2.0(Li et al., 2023c) as the annotator.

|  | Ability | Vicuna | Average |
|---|---|---|---|
| Random | 30.07 | 48.96 | 39.52 |
| Low-Consistency | 27.74 | 8.79 | 18.27 |
| High-Consistency | 30.44 | 60.81 | 45.63 |

**Diversity and Consistency** To assess the effect of combining quality and diversity in data selection, we conduct a set of experimental comparisons. Initially, we focus on quality alone in the selection process. Subsequently, we prioritize diversity in a separate selection process. In the final

| What are the main differences between Python and JavaScript programming languages? | |
| --- | --- |
| **High-Consistency** | **Low-Consistency** |
| The primary difference between Python and JavaScript is their syntax, data types, and use cases. Python is a high-level language with a simpler syntax than JavaScript which makes it easier to read and write code. It has fewer built-in data types compared to JavaScript, but Python provides more extensive library support for working with complex data structures such as lists and dictionaries. On the other hand, JavaScript is primarily used in client-side scripting and runs on the browser, while Python can be used both on the server side and client side. Additionally, Python supports multiple paradigms like object-oriented programming and functional programming, whereas JavaScript follows a single-paradigm approach of imperative programming. | Python is an interpreted language, while JavaScript is a compiled language. |
| **Write a symphony concert review, discussing the orchestra's performance and overall audience experience.** | |
| Last night I had the pleasure of attending a symphony concert featuring the Boston Symphony Orchestra at Tanglewood. The performance was truly remarkable as the musicians showcased their mastery over classical works by composers such as Beethoven, Mozart, and Brahms. The evening began with Beethoven's Fifth Symphony which set the tone for what would be an extraordinary evening of music making. Each instrument played its part flawlessly and created a wonderful harmony between them all. Following this, the orchestra performed pieces from other classical greats like Mozart and Brahms, bringing to life these beautiful melodies in ways that left me spellbound. Overall it was an incredible evening filled with captivating musical performances and I look forward to seeing more concerts put on by this amazing orchestra! | The symphony concert was nothing short of magical. The orchestra played with precision and emotion, transporting us to another world. The atmosphere in the venue was electric as the audience listened intently to every note. It was clear that everyone present was truly appreciating this incredible musical journey. |

Figure 5: A comparative analysis of model generation capabilities under two different selecting approaches is presented. The high-consistency method is represented on the left, while the low-consistency method is displayed on the right.

Table 4: The importance of combining quality and diversity(Llama2).

| | MMLU | Math | Code | Commonsense | World Knowledge | Average |
| --- | --- | --- | --- | --- | --- | --- |
| Alpaca-All | 47.93 | 13.12 | 13.41 | 55.04 | 20.83 | 30.07 |
| Diversity | 46.34 | 13.87 | 15.24 | 53.32 | 29.11 | 31.58 |
| Consistency | 45.26 | 15.39 | 15.85 | 44.80 | 30.94 | 30.45 |
| Diversity+Consistency | 47.12 | 15.69 | 15.85 | 56.51 | 29.83 | 33.00 |

phase, we integrate both quality and diversity in the selection. The outcomes are displayed in Table 4. The results suggest that a quality-centric approach may neglect data diversity, possibly constraining the model's proficiency in specific domains. Although a diversity-centric selection expands the data range, it risks incorporating lower-quality data, which could impair model performance. However, models that balance both quality and diversity in selection show enhanced performance in our tests. Quality guarantees that the model learns the interaction style of instructions, while diversity enables the model to master various styles, thereby improving its generalization and adaptability across different situations. Additional ablation experiments on Qwen2 are detailed in the Appendix B.

## 3.6 SELECTED DATA ANALYSIS

**Selection Reference** We perform an extensive analysis to probe the data selection biases across various models. Using GLM-4 Zeng et al. (2024), we categorized raw data into nine types and examined the filtering biases in different models, with key parameters provided in the Appendix D. The results in Table 5 show that models exhibit distinct data type preferences, with consistent selection patterns within model categories. This suggests the robustness of our method in tailoring data to model needs. Additionally, the Appendix D features a comparative analysis of selection biases across datasets.

**Data Diversity Analysis** To conduct an in-depth analysis of the data types our method typically selects and whether the chosen data maintains diversity, we employed the Self-instruct (Wang et al., 2023) to analyse. The findings are illustrated in Figure 6, indicating that the filtered dataset has en-

Table 5: Using GLM-4 to classify the data before and after selection. Here, $\Delta$ is calculated as $\frac{(\text{Alpaca-ALL} - \text{Selected})}{\text{Alpaca-ALL}}$.

| Category | Alpaca-ALL | Selected | $\Delta$ | Selected | $\Delta$ |
|---|---|---|---|---|---|
| Model | - | Llama2-7b | - | Qwen2-0.5b/1.5B | - |
| Discipline | 2193 | 242 | 88.96% | 277 / 283 | 87.37% / 87.10% |
| Language | 5855 | 72 | **98.77%** | 80 / 78 | **98.63% / 98.67%** |
| Knowledge | 15761 | 2012 | 87.23% | 2567 / 2537 | 83.71% / 83.90% |
| Comprehension | 3860 | 669 | **82.67%** | 767 / 817 | 80.13% / 78.83% |
| Reasoning | 837 | 94 | 88.77% | 118 / 89 | 85.90% / 89.37% |
| Creation | 12758 | 2103 | 83.51% | 2565 / 2780 | **79.89% / 78.20%** |
| Code | 626 | 59 | 90.58% | 82 / 90 | 86.90% / 85.62% |
| Mathematics | 3195 | 99 | 96.90% | 89 / 84 | 97.21% / 97.37% |
| Other | 5874 | 697 | 88.13% | 796 / 810 | 86.45% / 86.21% |

hanced task distribution while preserving the diversity present in the original data. More specifically, the filtered datasets exhibited a tendency to include creative and interpretive tasks such as "generate," "write," "create," "explain," and "describe," while tasks involving revisions such as "rewrite" and "edit" showed a relative decrease. This trend indicates that our selection approach is dedicated to enhancing data quality while also assuring a diversity of task types within the dataset. More analysis can be found in Appendix D.

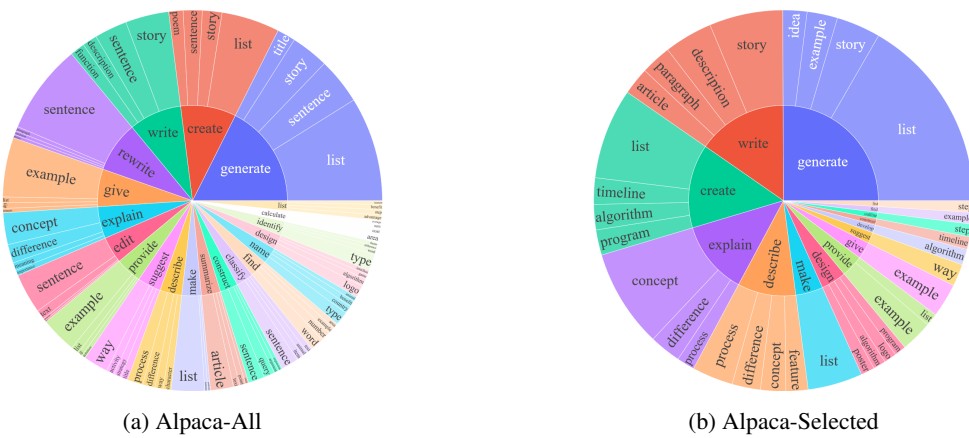

(a) Alpaca-All        (b) Alpaca-Selected

Figure 6: Comparing the diversity of instructions between the original alpaca source data and the filtered data involves analyzing the verb-noun structure of the instructions. The inner circle displays the top 20 most common root verbs found in the instructions, while the outer circle lists their corresponding first four direct noun objects. It is important to note that English commands come in various forms, and not all commands adhere strictly to this verb-noun structure. Therefore, the commands presented in this analysis only represent a portion of the total instructions.

## 4 RELATED WORK

**Instruction Dataset** Previous researches have concentrated on improving the model's instruction following ability using extensive instruction data sets (Ouyang et al., 2022; Chung et al., 2022). FLAN (Ouyang et al., 2022) effectively boosted model performance by transforming traditional NLP tasks into instruction datasets using instruction templates. Alpaca employs the self-instruct technique, utilizing advanced LLMs to generate a varied collection of 52k instructions (Taori et al.,

2023; Wang et al., 2023). Humpack (Li et al., 2023b) utilized an instruction reverse translation approach, generating training samples from seed models and improving model performance through self-filtering and iterative fine-tuning. WizardLm (Xu et al., 2023a) introduced an innovative method of progressively adjusting initial instructions to create more intricate instructions, thereby enhancing the performance of large language models. Additionally, Baize (Xu et al., 2023b) utilized powerful models to automatically produce multi-turn instructions and achieved commendable model performance through effective parameter adjustments. LIMA (Zhou et al., 2023) demonstrates that with just 1,000 meticulously curated high-quality data points, LLMs can exhibit significant improvements in command-following capabilities. This study demonstrates that even a small quantity of high-quality instruction data can lead to significant improvements in fine-tuning outcomes.

**Instruction Data Selection** Recent researchers have focused on minimizing the required data for instruction tuning, aiming to improve data efficiency and lower training costs. Intuitively, instruction mining (Cao et al., 2023) has established linear rules using specific natural language metrics for assessing the quality of instruction datasets. Furthermore, LLMs have shown remarkable language comprehension abilities, prompting researchers to also rely on other exceptional LLMs for assessing and selecting high-quality instruction data (Chen et al., 2023b). The AIT (Kung et al., 2023) proposes Prompt Uncertainty for filtering novel/informative instructions.Q2Q (Li et al., 2023a) uses a fine-tuned model to calculate the IFD index for each data point, which is then used to select high-quality data. In contrast, Self-Evolved (Wu et al., 2023) focuses on enhancing diversity through the utilization of the K-center method. MODS (Du et al., 2023) takes into account both data diversity and quality, but it still is restricted to relying on external models for quality assessment. Thus, we aim to assess the quality of individual data pieces within the pre-training model and integrate both quality and diversity to filter out high-quality instruction data.

## 5 CONCLUSION

Our approach merges the principles of quality and variety to refine the training dataset, enhancing instruction tuning. Initially, we assess the value of various data points by introducing noise, which helps us pinpoint the data that are most beneficial for model training. Subsequently, we broaden the dataset's reach while minimizing unneeded repetition, by boosting both the diversity between and within classes. Empirical evaluations across diverse datasets and models demonstrate that our innovative technique not only outstrips the performance achieved with full datasets but also notably exceeds the current state-of-the-art benchmarks. Our strategy not only decreases the resources necessary for training, but also significantly ameliorates model performance. Furthermore, it offers a fresh viewpoint on the utilization of noise in instruction tuning.

## 6 LIMITATION

Due to the limitations of computational resources, the largest model we use in our experiments is 7B, and we do not conduct experiments on larger models such as 70B. We do not perform an exhaustive gradient experiment to determine the optimal level of noise intensity. Furthermore, considering that different injection points may cause varying levels of interference, we do not explore the impact of different noise injection points on the experimental results.

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

## A  EXPERIMENT DETAILS

**Train Details**  We rent 4 NVIDIA A6000 for model training. During the training process, we adapt a full parameter fine-tuning strategy and utilized gradient accumulation techniques. Despite the fact that most of the instruction data is short, we still set the maximum data length to 4096 tokens. This setting does not affect our experimental results because the data padding is done according to the maximum length of the instructions in each batch. Our experiments are conducted based on the Alpaca instruction template shown in the Figure 7.

```
PROMPT_DICT = {
    "prompt_input": (
        "Below is an instruction that describes a task, paired with an input that provides
further context. "
        "Write a response that appropriately completes the request.\n\n"
        "### Instruction:\n{instruction}\n\n### Input:\n{input}\n\n### Response:"
    ),
    "prompt_no_input": (
        "Below is an instruction that describes a task. "
        "Write a response that appropriately completes the request.\n\n"
        "### Instruction:\n{instruction}\n\n### Response:"
    ),
}
```

Figure 7: The model training uses the following prompt template. During training, the correspond-ing instruction, input, and output are filled into their respective positions before being fed into the model.

**Dolly and Flan**  Instruction datasets are primarily categorized into three types: the first is gener-ated by advanced models, such as the Alpaca dataset; the second is manually written to ensure the quality of instructions; and the third converts traditional NLP datasets into instruction datasets using templates. Therefore, we have added the manually written Dolly dataset and the template-converted FLan dataset to validate the versatility and broad applicability of our method. In the experiments with the Dolly and Flan datasets, given the large size of the Flan dataset, which is challenging to fine-tune with limited resources, we randomly selected 15,000 pieces of data to match the size of the Dolly dataset. We used the same code to convert both datasets into a format suitable for the Alpaca model and conducted the training. For the multiple-choice question evaluation in the Flan dataset, since the model might not generate the corresponding options with precision, we used PPL (Perplexity) as the evaluation metric for the Flan dataset. Due to resource constraints, we did not test the screening of datasets of different sizes on the Flan and Dolly datasets, but instead, we only chose about 5,000 instructions for the experiment.

**Noise Injection**  After each piece of data is concatenated with the prompt template shown in the Figure 7, we inject noise parameters only in the region from instruction to input, while the other parts of the template remain undisturbed. In our main experiment, the injected Gaussian noise involves the configuration of two key parameters: mean and variance. Given that the information content of different instructions varies, it is clearly unreasonable to use fixed parameter values. Therefore,

we have adopted an adaptive parameter setting method. For each instruction, after embedding, we calculate the specific variance and mean of the region where noise is to be injected, and use these calculated values for initialization to achieve an appropriate semantic shift.

# B MORE ABLATION EXPERIMENTS

Within the Qwen2 architecture, we conduct a series of supplementary ablation studies to examine the influence of both inter-class and intra-class diversity on the model's efficacy. Experimental data in Table 6 shows that the concurrent preservation of both forms of diversity within the Qwen2-0.5B variant leads to a marked enhancement of the model's aggregate performance. Consistently, the Qwen2-1.5B model demonstrates peak performance when these diversity dimensions are retained. The empirical outcomes resoundingly validate the efficacy and preeminence of the diversity selection approach employed in our study.

Table 6: Additional Ablation Experiments on the Qwen2 Mode.

|  | Qwen2 | MMLU | Math | Code | Commonsense | World Knowledge | Average | Δ |
|---|---|---|---|---|---|---|---|---|
| Alpaca-all | 0.5B | 35.83 | 14.56 | 20.73 | 52.01 | 7.59 | 26.14 | — |
| Consistency +Inter-class(14%) | 0.5B | 36.19 | 35.41 | 23.17 | 52.61 | 6.68 | 30.87 | +4.73 |
| Consistency +inter-class +intra-class(14%) | 0.5B | 36.68 | 34.85 | 26.83 | 53.32 | 7.01 | 31.74 | **+5.60** |
| Alpaca-all | 1.5B | 50.47 | 39.73 | 33.54 | 69.94 | 13.77 | 41.19 | — |
| Consistency +Inter-class(14%) | 1.5B | 45.25 | 55.23 | 42.07 | 71.07 | 13.96 | 45.52 | +4.33 |
| Consistency +inter-class +intra-class(14%) | 1.5B | 45.10 | 57.54 | 40.24 | 71.25 | 14.16 | 45.66 | **+4.47** |

# C SELECTED EXAMPLES

```
{
    "instruction": "Develop an original story with the topic \"home\".",
    "input": "",
    "output": "Ivan was always told to appreciate the home that he was fortunate to have and he did. Despite his family\u2019s cramped and crowded living space, he was grateful for their roof, the safety it afforded, and the warmth it provided. All too often, though, this sense of home was fleeting. The stress of balancing work and school weighed on his parents, as did the knowledge of their growing debt each month. Ivan and his siblings often found themselves having to move from one rental home to another and at times, even double-up with his extended family. With each change, he grew to understand the different shapes of home. \n\nOne winter, the family stumbled upon a place that seemed different from the rest. At first, Ivan feared that the cold would seep in to their home and chill them all to their bones, but the windows and walls made it clear that this was a home built by people with a plan. Despite his own uncertainties, Ivan\u2019s excitement and sense of home quickly returned as he explored the familiar rooms and unknown corners. He could look through the kitchen window and see a mountain range in the distance and in the evening, he could lay in the backyard and count stars for hours. Here he felt something he hadn\u2019t felt before. He didn\u2019t have to leave, he belonged.\n\nHis parents soon shared his joy, feeling the comfort and support that came from having a place to call home. Months after their discovery of this place and over the course of many little and big victories, they were able to make the monthly payments and the house became their own.\n\nIvan and his family had finally found a home that felt like home."
},
```

Figure 8: Example: Develop an original story with the topic "home". (High-consistency)

```
{
    "instruction": "Write a short story where a character goes through a transformation",
    "input": "",
    "output": "Once upon a time, there was a young woman named Sarah who had it all. She had a successful career, wealthy friends, and a status that most aspired to. However, deep down Sarah was unfulfilled. She had a sense of emptiness that nothing seemed to quell. \n\nOne day, while out shopping, Sarah stumbled upon an old bookshop. Despite her reservation, she entered and was immediately taken by the musty aroma of ancient scrolls and leather-bound books. She began to browse, marveling at the sheer volume of knowledge contained within the walls of the shop. It was then, while engrossed in her exploration, that she discovered a book that changed her life. \n\nThe book contained an ancient scripture exhorting the reader to take control of their life. Sarah began to read, and as her eyes trailed along the writing she felt something stir within her. She began to question her life choices, her values, and her vision of the world. This led her to take a journey of self-discovery and inner transformation.\n\nSarah sacrificed the comforts she was accustomed to and began to build a life for herself that was true to her core. She discovered a set of values and beliefs which inspired her and led her to a life of service, compassion, and peace. Sarah had undergone a transformational experience of self-discovery and was determined to live a life that was truly aligned to her highest purpose. \n\nAnd so, with fingers burning with the passion of the journey, Sarah transformed and has never looked back since."
},
```

Figure 9: Example: Write a short story where a character goes through a transformation. (High-consistency)

```
{
    "instruction": "Write a short story where a character goes through a transformation",
    "input": "",
    "output": "Once upon a time, there was a young woman named Sarah who had it all. She had a successful career, wealthy friends, and a status that most aspired to. However, deep down Sarah was unfulfilled. She had a sense of emptiness that nothing seemed to quell. \n\nOne day, while out shopping, Sarah stumbled upon an old bookshop. Despite her reservation, she entered and was immediately taken by the musty aroma of ancient scrolls and leather-bound books. She began to browse, marveling at the sheer volume of knowledge contained within the walls of the shop. It was then, while engrossed in her exploration, that she discovered a book that changed her life. \n\nThe book contained an ancient scripture exhorting the reader to take control of their life. Sarah began to read, and as her eyes trailed along the writing she felt something stir within her. She began to question her life choices, her values, and her vision of the world. This led her to take a journey of self-discovery and inner transformation.\n\nSarah sacrificed the comforts she was accustomed to and began to build a life for herself that was true to her core. She discovered a set of values and beliefs which inspired her and led her to a life of service, compassion, and peace. Sarah had undergone a transformational experience of self-discovery and was determined to live a life that was truly aligned to her highest purpose. \n\nAnd so, with fingers burning with the passion of the journey, Sarah transformed and has never looked back since."
},
```

Figure 10: Example: Write an appropriate 500-word story about a surprise birthday party. (High-consistency)

```
{
    "instruction": "Simulate the output of a given code",
    "input": "public class HelloWorld {\n  public static void main(String[] args) {\n    System.out.println(\"Hello, World\");\n  }\n}",
    "output": "Hello, World"
},
{
    "instruction": "Extract the text from one paragraph of the following HTML document.",
    "input": "<html>\n  <head>\n    <title>Sample HTML page</title>\n  </head>\n  <body>\n    <h1>Title of the page</h1>\n    <p>This is a sample paragraph of text.</p>\n    <ul>\n      <li>List item 1</li>\n      <li>List item 2</li>\n    </ul>\n  </body>\n</html>",
    "output": "This is a sample paragraph of text."
},
{
    "instruction": "Clean up a set of customer reviews and rewrite some grammatically incorrect sentences. The output should be the fully edited customer reviews.",
    "input": "This place has the grateest food i have ever tasted. The costumer services was also ABSOLUTELY phenominal!",
    "output": "This place has the greatest food I have ever tasted. The customer service was also absolutely phenomenal!"
},
```

Figure 11: Examples of low consistency

```
{
    "instruction": "Guide the robot using the following grid.",
    "input":
"Grid:\n\n\u2554\u2550\u2550\u2550\u2566\u2550\u2550\u2550\u2566\u2550\u2550\u2550\u2557\n\u2551 X
\u2551  \u2551  \u2551\n\u2560\u2550\u2550\u2550\u256c\u2550\u2550\u2550\u256c\u2550\u2550\u2550\u2563
\n\u2551  \u2551  \u2551
\u2551\n\u2560\u2550\u2550\u2550\u256c\u2550\u2550\u2550\u256c\u2550\u2550\u2550\u2563 \n\u2551
\u2551  \u2551 Y
\u2551\n\u255a\u2550\u2550\u2550\u2569\u2550\u2550\u2550\u2569\u2550\u2550\u2550\u255d",
    "output": "Move up, move right, move right."
},
{
    "instruction": "Recognize the language in the given text.",
    "input": "\u4f5c\u4e3a\u4e00\u4e2a\u66fe\u7ecf\u7684\u82f1\u6587\u8001\u5e08",
    "output": "Chinese"
},
    "output": "Political Ad Spending Reaches Record Highs in U.S. Presidential Election Cycle of 2008."
},
{
    "instruction": "Classify this statement into \"Definitely true\", \"Possibly true\", \"Possibly false\", or \"Definitely
false\".",
    "input": "I eat pizza for lunch every day.",
    "output": "Possibly true"
},
```

Figure 12: Examples of low consistency

# D    DATA ANALYSIS

**Data type**    We conducted an in-depth analysis of the selection types for different data within the same model. It can be observed from the Table 7 that different models have different style preferences for different data. This may be due to the differences in the expression of instruction styles within different datasets, leading to varying data selection biases in the models.

Table 7: The tendency of data selection under different datasets with the same model. The quantities listed in the table are results after excluding some classification errors, such as when the model's output label is not within the range of 1-9.

| Category | Alpaca-ALL/Selected | $\Delta$ | Dolly-All/Selected | $\Delta$ |
|---|---|---|---|---|
| Model | Llama2-7b | - | Llama2-7b | - |
| Discipline | 2193/242 | 88.96% | 561 / 169 | 69.88% |
| Language | 5855/72 | **98.77%** | 113 / 36 | 68.14% |
| Knowledge | 15761/2012 | 87.23% | 10651 / 3639 | 65.83% |
| Comprehension | 3860 /669 | **82.67%** | 626 / 252 | 59.74% |
| Reasoning | 837/94 | 88.77% | 208 / 92 | **55.77%** |
| Creation | 12758/2103 | 83.51% | 856 / 339 | 60.40% |
| Code | 626 /59 | 90.58% | 5 / 1 | **80.00%** |
| Mathematics | 3195 / 99 | 96.90% | 162 / 53 | 67.28% |
| Other | 5874 / 697 | 88.13% | 1520 / 572 | 62.37% |

**GLM4**    During the process of invoking the GLM4 API, we use prompt words in the format shown in the Figure 13. We make sure to define each label in detail within the prompt words and provided a clear and intuitive example for each label category. When making the call, we combine these prompt words with the corresponding instructions from the dataset.

```
userprompt='Below are several types of instruction datasets. You need to determine which type the given instruction
belongs to based on the input, and output the corresponding number. Remember, do not provide any additional output.'
label="""
1. Discipline: Instructions typically involve knowledge in specific academic fields such as history, physics, chemistry, etc.
For example: "Explain Newton's three laws of motion."
2. Language: Instructions focus on the use of language, such as grammar, vocabulary, sentence structure, etc. For
example: "Please change the following sentence from a statement to a question."
3. Knowledge: Instructions require the provision of factual information or known data. For example: "List the top ten
highest mountains in the world."
4. Comprehension: Instructions necessitate explaining, summarizing, or elaborating on the understanding of a concept,
information, or text. For example: "Summarize the main idea of this article."
5. Reasoning: Instructions demand logical reasoning, analysis, or problem-solving. For example: "Based on these clues,
infer who the criminal is."
6. Creation: Instructions involve creative writing or expression, such as composing stories, poetry, scripts, or essays. For
example: "Write a short story about friendship."
7. Code: Instructions relate to programming and require writing, explaining, or modifying code. For example: "Write a
Python function to calculate the Fibonacci sequence."
8. Mathematics: Instructions involve mathematical calculations, problem-solving, or the application of mathematical
concepts. For example: "Solve this quadratic equation."
9. Other: Any instructions that do not fit into the above categories can be classified under this category. For example:
"Design a scientific experiment to test the reaction of plants to light."
"""
```

Figure 13: Related prompt for data classification using GLM-4. The specific explanations of our categories can also be seen from the figure.

**Morphological Feature Analysis**
We carefully analyzed the morphological characteristics of the filtered data, especially sequence length, to reveal the tendencies of our method in selecting data types. The results are shown in Table 8. Our approach tends to favor shorter sequences for instruction input and longer ones for output. This suggests that our technique leans towards selecting succinct, refined data for input instructions, while for output instructions, it chooses data offering comprehensive and detailed information. This strategic selection aids the model in concentrating on the crucial information during input processing, while offering ample and diverse information during output generation.

Table 8: Data length selected by different method.

|  | Input Length | Output Length | SUM |
|---|---|---|---|
| Alpaca-All | 83 | 270 | 353 |
| AlpacGasus | 73 | 339 | 412 |
| Ours | 57 | 530 | 587 |

| Verb | Noun | Alpaca_All | Verb | Noun | Selected | Verb | Noun | Deleted |
|---|---|---|---|---|---|---|---|---|
| generate | list | 859 | generate | list | 186 | rewrite | sentence | 741 |
| rewrite | sentence | 742 | explain | concept | 93 | generate | list | 673 |
| give | example | 489 | create | list | 86 | give | example | 457 |
| create | list | 480 | write | story | 70 | create | list | 394 |
| generate | sentence | 381 | make | list | 52 | sentence | sentence | 374 |
| write | story | 358 | write | description | 46 | write | story | 327 |

Table 9: Comparison of verb-noun pairs and their counts.

**Analysis of Verb-Noun** We conduct a more in-depth analysis of the instruction data and added specific gerund indicators. We count the top six gerunds in the Alpaca_all, Alpaca_selected, and Alpaca_deleted datasets. The results in Table 9 show that our method tends to select gerunds of the "generate" type construction, while almost completely excluding gerunds of the "rewrite" type. Intuitively, the generate-class data we have filtered is significantly superior to the rewrite type in terms of semantic richness. Generate-class data, born from creative thinking, is rich in detailed information, nuanced details, and innovative elements. In contrast, rewrite-class data appears more monotonous in terms of content information.

