# OpenReview forum: "Priority on High-Quality: Instruction Data Selection for Optimized Instruction Tuning"
_ICLR.cc/2025/Conference — Submitted to ICLR 2025_

### Official Review · Reviewer_WAh5 · 2024-10-28

**Soundness:** 2
**Presentation:** 2
**Contribution:** 2
**Rating:** 3
**Confidence:** 4

**Summary:**

This paper presents a data selection method which could define the quality of each instruction data and considering the balance between
data quality and data diversity. Experiments demonstrate that the proposed method maintain the performance of the whole dataset, and even outperforms the model trained on the full dataset.

**Strengths:**

1. Novel approach: The proposed method of using noise injection to identify the quality of instruction data is innovative and provides a new perspective in the field of instruction tuning.
2. Performance improvement: Experimental results show that the method significantly outperforms the model trained on the full dataset when using only 12% of the entire dataset, reducing training costs while improving model performance.
3. Consideration of quality and diversity: The study effectively combines data quality and diversity, addressing the limitations of existing methods that often focus on one aspect over the other.

**Weaknesses:**

1. This paper is not well-motivated.

> From this insight, we formulate a hypothesis: instructions that align with the knowledge absorbed during pre-training are more easily learned and integrated by the model through subsequent fine-tuning. We term these effective guiding instructions as "high-quality instructions. (at line 83)

There is no related publications or experiments (not mentioned in this paper) to support this opinion, but the method of this paper is motivated by such a non-verified 'idea'.

2. Lack of in-depth exploration of noise: The study did not conduct an exhaustive gradient experiment to determine the optimal level of noise intensity, leaving room for further investigation into the relationship between noise and data quality.
3. While the method shows promising results, the process of noise injection and how it exactly relates to data quality may lack interpretability. This could make it difficult for practitioners to understand and trust the method fully.
4. The method relies on the pre-trained model's behavior and certain assumptions such as the smoothness and clustering assumptions. If these assumptions do not hold true for certain datasets or models, the effectiveness of the method may be compromised.

**Questions:**

see the weakness part.

---

> ### Author Response · Authors · 2024-11-19
>
> Thanks for your comments and we reply to them below.
>
> + Our research methodology is grounded in the core hypothesis proposed by Lima [1]. Lima introduces the hypothesis that the knowledge of large models is predominantly acquired during the pre-training phase, while the fine-tuning process primarily involves learning to adhere to the style of specific instructions. This hypothesis, first posited by Lima, has garnered widespread acceptance in the field. Building upon this foundation, we introduced our own core hypothesis: instructions that align with the knowledge absorbed during pre-training are more readily learned and integrated by the model through subsequent fine-tuning. Our experimental results, as detailed in the paper, validate this hypothesis. The data selected by our method has significantly enhanced the model’s capabilities in areas such as code, mathematical, and commonsense.
>
> + Given the rationality of our method, we did not delve into detailed gradient experiments during the initial experimental phase. This was because our approach outperformed comprehensive data screening under noise intensities of 1 and 10. To further validate our findings, we supplemented gradient experiments. As the data in the table indicate, the experimental results show that our method surpasses the effects of comprehensive data training across different β values.
>
> |            |      MMLU | Math  | Code  | Commonsense | World Knowledge | Average |
> |:----------:|----------:|:-----:|:-----:|:-----------:|:---------------:|:-------:|
> | Alpaca_all |     47.93 | 13.12 | 13.41 |    55.04    |      20.83      |  30.07  |
> | ours(β=3)  |     46.13 | 11.90 | 15.24 |    53.48    |      26.59      |  30.67  |
> | ours(β=5)  |     45.28 | 14.10 | 17.07 |    51.76    |      28.86      |  31.41  |
> | ours(β=10) |     47.12 | 15.69 | 15.85 |    56.51    |      29.83      |  33.00  |
> | ours(β=15) |     43.43 | 13.57 | 15.24 |    53.89    |      28.37      |  30.90  |
>
>
> + The quality of the data itself is difficult to define in an intuitive manner. Existing research papers, which employ methods such as custom IFD scores or external model ratings, often fail to provide high-confidence interpretability. In contrast, our approach starts from the model's own perspective, using noise injection to evaluate whether the model has encountered related knowledge during the pre-training phase. Our hypothesis is that instructions consistent with the knowledge absorbed during pre-training are more easily learned and integrated by the model through subsequent fine-tuning. The experimental results confirm our hypothesis.
>
> + Firstly, the assumptions of smoothness and clustering form the foundational hypotheses in the domain of traditional machine learning, and their core concepts are fully realized in algorithms such as k-means clustering and the Consistency Regularization method. Secondly, we conducted separate experiments on three different types of instruction datasets (model-generated, manually written, and template-revised), and tested them on models of different types and with varying parameter scales to validate the effective generalization capability of our method.
>
> [1]Chunting Zhou, Pengfei Liu, Puxin Xu, Srinivasan Iyer, Jiao Sun, Yuning Mao, Xuezhe
> Ma, Avia Efrat, Ping Yu, Lili Yu, Susan Zhang, Gargi Ghosh, Mike Lewis, Luke Zettle-
> moyer, and Omer Levy. LIMA: less is more for alignment. In Alice Oh, Tristan Nau-
> mann, Amir Globerson, Kate Saenko, Moritz Hardt, and Sergey Levine (eds.), Advances
> in Neural Information Processing Systems 36: Annual Conference on Neural Informa-
> tion Processing Systems 2023, NeurIPS 2023, New Orleans, LA, USA, December 10 - 16,
> 2023, 2023.

---

### Official Review · Reviewer_cwnb · 2024-10-29

**Soundness:** 3
**Presentation:** 3
**Contribution:** 3
**Rating:** 6
**Confidence:** 3

**Summary:**

This paper proposes a method that utilizes noise injection to identify the quality of instruction data. They also implement the strategy of combining inter-class diversity and intra-class diversity to select instruction data accompanied by the quality identification method. Experimental results demonstrate that the method outperforms some previous methods and outperforms the model trained on the full dataset when utilizing a small percentage of the entire dataset.

**Strengths:**

1. It is reasonable to use noise inject to judge the quality of instruction data. The authors have explained their insight in the paper, and it is also effective in experiments.
2. The writing is clear and easy to understand.
3. The ablation experiment well demonstrated the roles of consistency and diversity in the method.

**Weaknesses:**

1. Only one data set is used for the experiment. Using multiple data can illustrate the generality of the method. For example, Mods[1] also uses a larger mixture instruction datasets which is composed of instruction data from several different datasets.
2. I noticed that the methods cited and compared in this paper are up to 2023, but there are still new methods that have not been compared in related works or experiments, e.g. [2][3].

[1] MoDS: Model-oriented Data Selection for Instruction Tuning

[2] Superfiltering: Weak-to-Strong Data Filtering for Fast Instruction-Tuning

[3] LESS: Selecting Influential Data for Targeted Instruction Tuning

**Questions:**

See weaknesses

---

> ### Author Response · Authors · 2024-11-19
>
> Thanks for your comments and we reply to them below.
>
> + We appreciate your positive feedback on our paper. However, concerning the use of a larger dataset, it is challenging to undertake this task during the rebuttal period, as the training and evaluation of models is a time-consuming and resource-intensive process. In our study, we have chosen the widely recognized alpaca dataset as the primary subject for our experiments, which has been used as a key dataset in numerous studies on instruction fine-tuning and selection. Moreover, we have also achieved excellent results on the smaller dolly-15k dataset, which further attests to the effectiveness of our method across datasets of different sizes.
>
> + The contribution of the LESS paper is primarily reflected in the dataset selection for specific target tasks, which does not entirely align with our multi-task screening approach. Considering the constraints of computational resources and time, we only supplemented the experiments from the Superfiltering paper. We utilized the publicly available dataset from that paper and selected a similar number of data points as our main experiment. The results presented in the following table indicate that our method outperforms the approach described in the Superfiltering paper across all noise intensities. In our experiments, the method proposed in the LESS paper did not perform ideally, which may be due to its data selection method not fully considering the model's specific data needs. Relying solely on an external model to select data may not be sufficient to choose the most suitable data for the model itself.
>
> |                   |      MMLU | Math  | Code  | Commonsense | World Knowledge | Average |
> |:-----------------:|----------:|:-----:|:-----:|:-----------:|:---------------:|:-------:|
> | Superfiltering    |     41.03 | 7.73  | 11.59 |    49.63    |      19.14      |  25.82  |
> |     ours(β=3)     |     46.13 | 11.90 | 15.24 |    53.48    |      26.59      |  30.67  |
> |     ours(β=5)     |     45.28 | 14.10 | 17.07 |    51.76    |      28.86      |  31.41  |
> |    ours(β=10)     |     47.12 | 15.69 | 15.85 |    56.51    |      29.83      |  33.00  |
> |    ours(β=15)     |     43.43 | 13.57 | 15.24 |    53.89    |      28.37      |  30.90  |

---

### Official Review · Reviewer_dfXN · 2024-10-31

**Soundness:** 2
**Presentation:** 3
**Contribution:** 2
**Rating:** 5
**Confidence:** 3

**Summary:**

This paper introduces a novel approach for selecting high-quality instructional data for tuning LLMs. Unlike previous research, which primarily emphasizes the volume of instructional data, this work addresses limitations in defining and balancing the quality and diversity of the data. The authors propose a method that uses noise injection to identify high-quality instructional data, analyzing the pre-trained model’s response to perturbed inputs to measure the consistency of probability distributions. This approach enables a model-centric quality assessment without relying on external scoring models. Additionally, the framework integrates both inter-class and intra-class diversity to ensure comprehensive task coverage and reduce data redundancy. Experimental results demonstrate that the proposed method achieves strong performance on unfiltered datasets across various standard benchmarks.

**Strengths:**

- This paper is well-written in general and well-motivated.
- The research direction of studying how to select datasets for LLM supervised fine-tuning is of practical importance.
- The proposed data selection pipeline is easy to follow and intuive.
- The experimental results of the proposed data selection pipeline seem to be very promising.

**Weaknesses:**

- The paper lacks fundamental explanation and justification for the assumption that low KL divergence after noise perturbation indicates high data quality. It is unclear whether this is purely heuristic or based on a specific hypothesis or experimental observation.
- The proposed method seems computationally intensive. For instance, all the perturbed data must pass through the LLM to obtain predictions and calculate KL divergence.
- There is no comparison between the proposed data selection pipeline and similar methods, e.g., [1].
- It is unclear whether the dataset selected using one LLM (preferably at a smaller scale, e.g., 7B) can be broadly applied to fine-tune various LLMs.

[1] https://arxiv.org/abs/2405.00705

**Questions:**

- Why is low KL divergence among LLM outputs before and after noise perturbation an indicator of high data quality? Is there any fundamental reason behind it?
- Are there potential methods to accelerate the speed of data selection in the proposed approach?
- Could one use, say, Llama3 7B to select data and then use it for fine-tuning Llama3 405B?
- How does the proposed method compare to the approach presented in https://arxiv.org/abs/2405.00705?

---

> ### Author Response · Authors · 2024-11-19
>
> Thanks for your comments and we reply to them below.
>
> **Response to weaknesses**:
> + Our initial concept draws from traditional machine learning, where data points with different labels are separated in low-density regions, and similar data points yield similar outputs. Consequently, applying actual perturbations to learned data should not result in significant changes to predictions, indicating consistency in the outputs. In line with the assumptions proposed in Lima’s paper, we hypothesize that the knowledge underlying instructions is consistent with the knowledge acquired during pre-training, making it easier for models to learn the style of instructions. Intuitively, once knowledge is learned, the model can quickly adapt to a new style following a certain degree of stylistic rewriting.
>
> + The core idea of our paper is that the best data for a model is that which suits the model itself. Therefore, unlike traditional selection methods that employ external models for data filtering, we use the model itself for data selection, which may entail some resource overhead.
>
> + In our paper, we have compared our selection method with others. For instance, AlpaGasus [1] employs an external model to score each data point, using these scores to assess the quality.
>
> + In addition to the llama2 model, our paper extends the analysis to include the qwen2-0.5B and qwen2-1.5B models to validate the generalizability of our method across different model types and sizes.
>
> **Response to questions**:
>
> + Q1: Our core hypothesis is that instructions consistent with knowledge absorbed during pre-training are more easily learned and integrated by the model through subsequent fine-tuning. Therefore, by introducing data perturbations and comparing the consistency of output probability distributions, we verify whether the model has learned the data during pre-training. The experimental results confirm our hypothesis, with the selected data enhancing the model’s capabilities in areas such as code and mathematics.
>
> + Q2 and Q3: We conducted an in-depth analysis of the filtered data by using GLM4 to classify texts into 10 categories. We observed that different models show certain discrepancies in their preferences for data selection, both in terms of the most and least favored data. Within the same class of models, data selection trends are consistent, which may be attributed to the similar training data used during pre-training. Based on this, we propose that data filtered by smaller models can be utilized to train larger models. This approach could potentially be an effective method to reduce resource expenditure in the future.
>
> |               | Alpaca-ALL | Alpaca_Selected |  Rate_of_Change   | Alpaca_Selected |     Rate_of_Change    |
> |:-------------:|:----------:|:---------------:|:--------:|:---------------:|:-----------:|
> |   Model       |     -      |    LLama2-7b    |     -      | Qwen2-0.5b/1.5B |            -            |
> |  Discipline   |    2193    |       242       |   88.96%   |    277 / 283    |     87.37% / 87.10%     |
> |   Language    |    5855    |       72        | **98.77%** |     80 / 78     | **98.63%** / **98.67%** |
> |   Knowledge   |   15761    |      2012       |   87.23%   |   2567 / 2537   |     83.71% / 83.90%     |
> | Comprehension |    3860    |       669       | **82.67%** |    767 / 817    |     80.13% / 78.83%     |
> |   Reasoning   |    837     |       94        |   88.77%   |    118 / 89     |    85.90% /  89.37%     |
> |   Creation    |   12758    |      2103       |   83.51%   |   2565 / 2780   | **79.89%** / **78.20%** |
> |     Code      |    626     |       59        |   90.58%   |     82 / 90     |     86.90% / 85.62%     |
> |  Mathematics  |    3195    |       99        |   96.90%   |     89 / 84     |     97.21% / 97.37%     |
> |     Other     |    5874    |       697       |   88.13%   |    796 / 810    |     86.45% / 86.21%     |
>
> + Q4: This reference literature does not have corresponding open-source code, making it difficult to accurately reproduce the code from the paper in a short period of time. In our paper, we have compared our selection method with others. For instance, AlpaGasus [1] employs an external model to score each data point, using these scores to assess the quality.
>
> [1] L. Chen, S. Li, J. Yan, H. Wang, K. Gunaratna, V. Yadav, Z. Tang, V. Srinivasan, T. Zhou, 306 H. Huang, and H. Jin. Alpagasus: Training a better alpaca with fewer data. In The Twelfth 307 International Conference on Learning Representations, ICLR 2024, Vienna, Austria, May 7-11, 308 2024.

---

### Official Review · Reviewer_g9bs · 2024-11-04

**Soundness:** 3
**Presentation:** 2
**Contribution:** 2
**Rating:** 5
**Confidence:** 3

**Summary:**

The authors propose a data selection approach using noise injection to assess instruction quality by introducing controlled noise into the input data and measuring the consistency of the model's output distribution. This method identifies high-quality data that aligns with the model’s pre-trained knowledge, allowing for more efficient fine-tuning. Additionally, to prevent over-representation of certain data types, the authors implement inter-class and intra-class diversity strategies using clustering and cosine similarity to ensure a balanced dataset. Their approach demonstrates superior model performance using only 12% of the original dataset, reducing training costs while enhancing model efficiency. As part of their contributions, the authors also publish a high-quality instruction dataset curated through their method, offering a valuable resource for further research in instruction tuning.

**Strengths:**

- The approach of assessing data quality via noise injection is innovative, presenting a fresh perspective on defining quality in instruction tuning without relying on external scoring models. This originality extends to combining noise-based quality assessment with diversity strategies, addressing a critical gap in current methods that either prioritize quality or diversity but rarely balance both effectively.
- the use of inter-class and intra-class diversity strategies showcases a nuanced understanding of dataset composition and further enhances the quality of the work.
- The explanation of noise injection and consistency measures is detailed and accessible, allowing readers to understand the novel concepts and replicate the approach.

**Weaknesses:**

- The paper’s experimental results could be improved with additional baseline comparisons. While the authors compare their method to full-data training, adding comparisons to recent quality and diversity-oriented methods would allow readers to better gauge the benefits and trade-offs of this approach in a broader context.
- The paper solely relies on output distribution consistency following noise injection to define quality, which may overlook other nuanced aspects of instruction quality, such as contextual relevance, instruction clarity, or alignment with specific model objectives.
- Although the paper introduces inter-class and intra-class diversity strategies, it only uses k-means clustering and cosine similarity to ensure diversity. This limited approach might lead to clusters that fail to represent complex hierarchical or nuanced differences within data types.

**Questions:**

- The paper’s noise injection approach is innovative, but it lacks an in-depth exploration of the noise parameters. Could you provide more details on how the noise injection parameters, such as the scaling factor β, were selected? Did you experiment with different values, and if so, how did these affect the results?

- The selected high-quality data are determined based on consistency following noise injection, but it would be helpful to understand what specific characteristics these data share. Could you analyze or describe typical examples of the high-consistency data and explain why these instructions align well with model knowledge?

---

> ### Author Response · Authors · 2024-11-19
>
> Thanks for your comments and we reply to them below.
>
> **Response to Q1**:
> 	In our paper, we primarily present the results of data filtering with the parameter β set to 1 and 10. To further refine our research, we supplement the experimental results for different β values here. Through comparative analysis, we find that our selection method consistently outperforms the full-data training across different β values.
>
> |            |      MMLU | Math  | Code  | Commonsense | World Knowledge | Average |
> |:----------:|----------:|:-----:|:-----:|:-----------:|:---------------:|:-------:|
> | Alpaca_all |     47.93 | 13.12 | 13.41 |    55.04    |      20.83      |  30.07  |
> | ours(β=3)  |     46.13 | 11.90 | 15.24 |    53.48    |      26.59      |  30.67  |
> | ours(β=5)  |     45.28 | 14.10 | 17.07 |    51.76    |      28.86      |  31.41  |
> | ours(β=10) |     47.12 | 15.69 | 15.85 |    56.51    |      29.83      |  33.00  |
> | ours(β=15) |     43.43 | 13.57 | 15.24 |    53.89    |      28.37      |  30.90  |
>
> **Response to Q2**:
> 	We extract gerunds from the Alpaca_all, Alpaca_selected, and Alpaca_deleted datasets and present the top six results in terms of frequency, as shown in the table below. In analyzing the characteristic features of high-consistency data, we find that these samples tend to include gerund structures of the “generate” type, while gerunds of the “rewrite” type are seldom selected. The knowledge content embedded in these generative instructions is significantly greater than that in rewrite instructions, and the intrinsic knowledge of such instructions is more likely to align well with what the model has learned. Furthermore, we analyze the instruction length in data selection within our paper and discover that our method tends to favor longer instructions. This is similar to the type of data the model is exposed to during the pre-training phase, where long data is predominantly handled. This preference for longer instructions may also be an indication of data consistency, as longer instructions typically contain more information.
>
> |    Verb    |   Noun   | count(Alpaca_All)  |   Verb   |    Noun     | count(Alpaca_Selected_LLama2) |   Verb   |   Noun   | count(Alpaca_Deleted_LLama2) |
> |:----------:|:--------:|:------------------:|:--------:|:-----------:|:-----------------------------:|:--------:|:--------:|:----------------------------:|
> |  generate  |   list   |        859         | generate |    list     |              186              | rewrite  | sentense |             741              |
> |  rewrite   | sentence |        742         | explain  |   concept   |              93               | generate |   list   |             673              |
> |    give    | example  |        489         |  create  |    list     |              86               |   give   | example  |             457              |
> |   create   |   list   |        480         |  write   |    story    |              70               |  create  |   list   |             394              |
> |  generate  | sentense |        381         |   make   |    list     |              52               | sentence | sentence |             374              |
> |   write    |  story   |        358         |  write   | description |              46               |  write   |  story   |             327              |

---

### Meta-Review · Area_Chair_knTu · 2024-12-12

**Metareview:**

This paper focuses on selecting critical instruction data for optimizing instruction tuning. It proposes utilizing noise injection to identify the quality of instruction data and consider both the quality and diversity of selected data. Experiments demonstrate the effectiveness of the proposed method. A high-quality instruction set is curated through the proposed method and published, which contributes to the research community.

The strengths of this paper are reflected in several aspects. First, the research problem and direction of selecting more important instruction data are valuable and promising. Second, the motivation for considering both quality and diversity is overall clear. Third, the reported experimental results are good. The weaknesses mainly lie in two perspectives. There is a lack of an intuitive and in-depth analysis of the effectiveness and technical contributions of the proposed method. Besides, the effectiveness and adaptability of the proposed method in realistic scenarios (such as the validity of underlying assumptions) are concerned.  The weaknesses outweigh the strengths, which puts this work below the acceptance line.

**Additional Comments On Reviewer Discussion:**

Before rebuttal, four reviewers point out their concerns from several aspects.

- The motivation of this paper is not strong and not clear (reviewer WAh5).
- The experiments are not sufficient to verify the superiority of the proposed method and justify the claims (reviewers g9bs and dfXN).
- The mechanism and in-depth analysis of the proposed method are not clear and unconvincing (reviewers g9bs, dfXN, and WAh5).

The feedbacks provided by the authors mainly address the problem in experiments, by providing more empirical results. However, it does not well answer the noise mechanism in instruction data selection, how to formally connect the noise to high quality, and the reasonableness of introduced assumptions.

Most of the reviewers are negative about the current form of this paper. Besides, no one champions it. Based on the above, AC then makes the final rejection recommendation.

---

### Decision · Program_Chairs · 2025-01-22

Reject